# Effect of Tannic Acid on the Corrosion Behavior of W18Cr4V in a Simulated Wood Environment and Its Inhibition Mechanism

Na Jia [ID], Chunxin Wang [ID], Chunpu Zhang and Jiuqing Liu *

College of Mechanical and Electrical Engineering, Northeast Forestry University, Harbin 150040, China; jiana@nefu.edu.cn (N.J.); wcx2000@nefu.edu.cn (C.W.); 2021111712@nefu.edu.cn (C.Z.)
* Correspondence: jiuqingliu@nefu.edu.cn

**Abstract:** This study explores the effect of tannic acid on the corrosion of woodworking tool material W18Cr4V in a simulated wood environment. The weight loss method, potentiodynamic polarization, and electrochemical impedance spectroscopy were used to study the corrosion and the corrosion inhibition process of W18Cr4V in an environment of acetic acid solution with different concentrations of tannic acid. The results show that after continuous immersion for 20 h, low concentrations (1 wt% and 3 wt%) of tannic acid promoted the corrosion of W18Cr4V, while high concentrations (5 wt% and above) of tannic acid had a significant anti-corrosion effect, with a corrosion inhibition efficiency of about 64% for 10 wt% tannic acid. Scanning electron microscopy, energy-dispersive X-ray spectroscopy, and X-ray photoelectron spectroscopy were used to further verify and elucidate the inhibition mechanism. It was found that tannic acid can form a dense and effective corrosion inhibition film composed of iron–tannin complexes on the surface of W18Cr4V. This study not only provides a new perspective on understanding the corrosion effect of tannic acid on woodworking tools but also offers new insights for developing effective bio-corrosion protection strategies.

**Keywords:** corrosion; wood; tannic acid; acetic acid; W18Cr4V

## 1. Introduction

Wood, as a widely used natural resource, plays a crucial role in various fields. However, during the wood processing, the wear of woodworking tools has been a concern as it directly affects production efficiency and may reduce the final product quality. In this context, it is essential to delve into the nature of tool wear, which is more complex than it appears on the surface. The traditional view is that the wear of woodworking tools is mainly caused by mechanical action, but recent research has proposed a new perspective: the degree of tool wear cannot be fully explained by the mechanical properties of wood or differences in processing parameters alone [1]. In fact, in addition to continuous mechanical stress, the chemical components in wood may also have a significant impact on tool wear [2,3].

Research has found that a piece of wood may contain more than 700 different extracts [4–6]. However, only three types of extracts have been proven to cause corrosion to metals in contact with wood or wood sap. They are small organic acids (such as acetic acid and formic acid), tannic acid (or other polyphenols), and phenols with two or three adjacent hydroxyl groups (such as catechol and pyrogallol) [7]. However, the presence of catechol (1,2-dihydroxybenzene) and pyrogallol (1,2,3-trihydroxybenzene) is not common in solid wood cutting, as they are mainly produced during the pulping process due to the decomposition of lignin [8–10]. Therefore, in the cutting and processing of solid wood, more attention should be paid to the impact of small organic acids and tannic acid on metal wear corrosion.

In fact, corrosion wear and mechanical wear are not independent of each other. They often have a coupled effect, accelerating the wear process of the tool and reducing its service life. The moisture content of wood, pH value, acid content, and the temperature of the processing environment may all affect the corrosion behavior of tools during processing [11].

It is worth noting that corrosion may not only occur during the cutting process but may also occur during downtime due to the residue of wood extracts on the tool surface [11,12]. Therefore, this study focuses on acetic acid and tannic acid, two common chemical components of wood, aiming to reveal their specific roles and impact mechanisms in the tool corrosion wear process.

## 2. Research Background

### 2.1. Acetic Acid

Studies have shown that the water-soluble extracts of most woods have acidic properties, with pH values typically ranging from 2.9 to 5.5 [13–15]. Among all the extracts, acetic acid and formic acid are almost the most important factors affecting the pH value of wood, with acetic acid being particularly prominent. This is mainly because the hydrolysis of acetates inside the wood releases acetic acid, as is shown in the following:

$$R - OCOCH_3 + H_2O \rightleftharpoons R - OH + CH_3COOH$$

Glucoside acetate acetic acid.

There is a dynamic equilibrium in this reaction, making the moisture in the wood always acidic. Due to the volatility of acetic acid, it evaporates from the equilibrium system, stimulating the continuous formation of acetic acid.

The content of acetate in wood is generally between 1%–6 wt%, with hardwoods typically having higher contents than softwoods [16]. Additionally, the acetyl and formyl groups in hemicellulose can also form acetic acid and formic acid through hydrolysis, with acetic acid accounting for over 90% of the formation [17–19]. Hemicellulose typically makes up about 7%–25 wt% of wood [20]. Therefore, in the case of complete hydrolysis of cellulose, the acetic acid content in the wood is 7.3%–28.5 wt%. This hydrolysis process is chemical, not biological.

Therefore, during wet wood processing, such as peeling, rotary cutting, planing veneers, and defibrating, acetic acid is one of the most important factors in the corrosion wear of woodworking tools.

### 2.2. Tannic Acid

Tannic acids (chemical formula $C_{76}H_{52}O_{46}$) are a class of polyphenols present in plants, generally considered to be the combination of five hydroxyl groups of glucose and galloyl groups. They are distributed in bark, leaves, cones, roots, and stems [21], such as the bark of *quercus* and *acacia* and the heartwood of *pinus* and *eucalyptus robusta* [19,22]. The adjacent hydroxyl groups in tannic acid can interact with metal ions to form water-insoluble metal tannates [23,24], which turn into a deep blue/purple precipitate [25]. However, there is still significant controversy regarding the effect of this chelation on metal corrosion.

In studies exploring the impact of tannic acid on metal corrosion in solid wood, most research indicates that tannic acid enhances metal corrosion. Winkelmann, H. et al. believe that tannic acid increases the acidity of the solution, thereby promoting metal corrosion [26,27]. Pugsley, V.A. et al. studied the effect of tannic acid on the fatigue performance of hard alloys. The results showed that tannic acid has a corrosive effect on the specimens and has a synergistic effect with cyclic fatigue loads [28]. Pugsley, V.A. et al. also studied the role of tannin in saw blade corrosion and found that the strength of all steel samples decreased with prolonged exposure, and some alloys exhibited stress corrosion cracking [29].

However, in other research areas, tannic acid is often used as a corrosion-resistant coating or a corrosion inhibitor to protect metals. For example, Adam, M.R. et al. extracted tannic acid from *mangrove* and found that it had a high corrosion inhibition effect on steel soaked in salt solution [30]. Usman, B.J. et al. studied the inhibitory effect of tannic acid on X60 steel in a salt solution and found that tannic acid had a significant inhibitory effect on X60 steel [31]. Qian, B. et al. found that tannic acid forms a corrosion inhibition film, which

is relatively stable in the dry–wet cycle of seawater on the steel surface, thereby achieving corrosion protection for low-carbon steel [32]. Xu, W.H. et al. found that the corrosion rate of Q235 steel soaked in a salt solution decreased as the concentration of added tannic acid increased [33].

When studying the impact of tannic acid on metal corrosion, it was found that its effects showed significant differences in different research backgrounds. This difference is likely due to the different environmental conditions used in each study. For example, during the processing of solid wood, due to the effect of mechanical friction, the metal tannates formed by tannic acid may peel off from the metal surface, triggering a continuous chelation reaction and causing metal corrosion. However, in other environments, metal tannates may form a protective film on the metal surface, effectively inhibiting the process of metal corrosion.

It is worth noting that in the most common acidic environment in wood—the acetic acid environment—there is still a lack of research on the impact of tannic acid on tool corrosion. This area needs further exploration to fill the existing research gap. The main motivation for proposing this research topic is that acetic acid and tannic acid are the most common and important acidic substances in wood. As commonly used tools in solid wood processing, it is of great significance for users to understand the mechanism of these two acids on tool corrosion.

### 2.3. Purpose of This Study

In this study, we use acetic acid to simulate the wood environment in order to deeply explore the impact of tannic acid on the corrosion behavior of W18Cr4V steel in a wood environment and reveal the corrosion mechanism at the microscopic level.

Firstly, we are committed to studying the corrosion effects of acetic acid and tannic acid at different concentrations on W18Cr4V steel in order to reveal the relationship between corrosion behavior and acid concentration through comparative analysis. Additionally, we also focus on corrosion behavior in a mixed acid environment, providing a theoretical basis for studying the complex chemical environment that may be encountered in the actual wood processing environment.

Secondly, we use advanced surface analysis techniques such as scanning electron microscopy (SEM), energy-dispersive X-ray spectroscopy (EDS), and X-ray photoelectron spectroscopy (XPS) to conduct a detailed microscopic structural analysis of the corroded surface of W18Cr4V steel. We hope to gain a deeper understanding of the composition, structure, and impact of the products formed during the corrosion process through these techniques.

Lastly, we study and explain how tannic acid inhibits the corrosion of W18Cr4V steel in an acetic acid environment from both theoretical and experimental perspectives. The results of this study are expected to provide a deeper understanding of the impact of organic acids on metal corrosion, which can be useful for understanding and utilizing it.

In summary, the goal of this study is to deeply research the corrosion mechanism of W18Cr4V steel by tannic acid in a simulated wood environment. By understanding this mechanism, we aim to develop more effective anti-corrosion strategies, select more appropriate tool materials, and improve the tool lifespan and processing efficiency during wood processing. The research is expected to provide a scientific basis for the development of the woodworking machinery industry, with the ultimate goal of making significant contributions to the field.

### 3. Experiment
### 3.1. Research Materials

In the present study, we employed a conventional three-electrode system for electrochemical measurements. The working electrode (WE) was fabricated from W18Cr4V, a tungsten-based high-speed steel commonly used in solid wood processing, supplied by Chuli Tools Co., Ltd. (Lishui, Zhejiang Province, China). The detailed composition

of this material is presented in Table 1. For the counter electrode (CE), a platinum electrode manufactured by Shanghai Yueci Electronic Technology Co., Ltd. (Shanghai, China) was chosen, with dimensions of 10 mm × 10 mm × 0.2 mm. The reference electrode (RE) utilized was a saturated Ag/AgCl electrode, also provided by Shanghai Yueci Electronic Technology Co., Ltd. (Shanghai, China), possessing a standard electrode potential of +0.199 V at 25 °C.

**Table 1.** Composition of W18Cr4V working electrode.

| C (wt%) | Mn (wt%) | Si (wt%) | S (wt%) | P (wt%) | Cr (wt%) | V (wt%) | W (wt%) | Fe (wt%) |
|---|---|---|---|---|---|---|---|---|
| 0.73~0.87 | 0.10~0.40 | 0.20~0.40 | ≤0.030 | ≤0.030 | 3.80~4.50 | 1.00~1.20 | 17.20~18.70 | Residual |

The chosen working electrode material was cut into square samples of 10 mm × 10 mm × 3 mm. After connecting the wire, it was fixed with epoxy resin, sealing other surfaces to ensure that only the 10 mm × 10 mm surface was exposed for the corrosion experiment. The surface was ground using 400, 600, 800, 1200, and 1500 grit sandpaper and then polished with a polishing machine. It was then cleaned sequentially with acetone and anhydrous ethanol.

The corrosive medium was a solution based on 10 wt% acetic acid (AA), with different concentrations (0 wt%, 1 wt%, 3 wt%, 5 wt%, 8 wt%, 10 wt%) of tannic acid. Both acetic acid and tannic acid used in this study were of analytical grade and were sourced, respectively, from Jiangsu Qiangsheng Functional Chemistry Co., Ltd. (SuZhou, Jiangsu Province, China) and Changde Beikeman Biotechnology Co., Ltd. (Changde, Hunan Province, China). The aim of this research is to investigate in-depth the specific influence of varying concentrations of tannic acid on the corrosion behavior of W18Cr4V material under simulated wood conditions.

*3.2. Weight Loss Method*

The weight loss method is a commonly used method to evaluate the corrosion rate of materials. The corrosion rate ($v_r$) is calculated according to Equation (1).

$$v_r = \frac{\Delta W \times 87600}{A \times t \times \rho},\tag{1}$$

wherein $\Delta W$ represents the mass loss of W18Cr4V soaked under the same conditions, with the unit being g. $A$ represents the exposed area, with the unit being cm$^2$. In this experiment, $A = 1$ cm$^2$. t represents the soaking time of the sample in the corrosive medium (measured in hours or days); in this experiment $t = 20$ h. Referring to the GB/T 9943-2008 standard [34], for W18Cr4V, $\rho = 8.5$ g/cm$^3$. The efficiency of corrosion is calculated using Equation (2).

$$\eta_W(\%) = \frac{v_0 - v_i}{v_0} \times 100\%\tag{2}$$

where $v_0$ represents the corrosion rate of the specimen under conditions without TA while $v_i$ represents the corrosion rate of the specimen under different concentrations of TA [35].

The samples were initially measured using an analytical balance (accuracy ± 0.1 mg) and then immersed in the prepared solutions with different concentrations of tannic acid for 20 consecutive hours. After immersion, the samples were first cleaned using a solution composed of 20 wt% hydrochloric acid + 5 wt% methylamine to remove corrosion products [36], then with acetone and anhydrous ethanol. After cleaning and drying the samples, they were weighed again. Each experimental condition was repeated three times to obtain the average weight of the samples before and after corrosion.

*3.3. Electrochemical Testing*

3.3.1. Potentiodynamic Polarization (PDP) Test

The PDP test mainly examines the kinetic process of corrosion reactions. The corrosion potential ($E_{corr}$) and corrosion current density ($I_{corr}$) are obtained using the Tafel extrapolation method. The category of the inhibitor, as well as the inhibition efficiency, can be thereby acquired. If $E_{corr}$ > 85 mV, the inhibitor might be anodic or cathodic (relative to the blank solution). If $E_{corr}$ < 85 mV, the inhibitor is of mixed type [37]. The corrosion or inhibition efficiency ($\eta_P$) is calculated according to Equation (3) [38].

$$\eta_P(\%) = \frac{I_{0corr} - I_{corr}}{I_{0corr}} \times 100\%, \tag{3}$$

$I_{0corr}$ represents the corrosion current density of the electrode under conditions without TA. $I_{corr}$ represents the corrosion current density of the electrode under different TA concentrations.

This experiment used CS350H for PDP testing. After the samples were immersed for 20 h, they were tested. The scanning rate was 3 mV/s, and the scanning range was open circuit potential (OCP) ± 500 mV.

3.3.2. Electrochemical Impedance Spectrum (EIS) Test

EIS (electrochemical impedance spectroscopy) testing technology is a non-destructive corrosion measurement technique. By analyzing the EIS and the appropriate equivalent circuit, parameters such as solution resistance ($R_s$), charge transfer resistance ($R_{ct}$), double-layer capacitance ($C_{dl}$), double-layer film capacitance ($C_c$), Warburg resistance ($W$), etc., can be obtained [39,40]. The corrosion or inhibition efficiency ($\eta_E$) is further calculated using Formula (4) [41,42].

$$\eta_E(\%) = \frac{R_{ct}^0 - R_{ct}}{R_{ct}^0} \times 100\%, \tag{4}$$

In Formula (4), $R_{ct}^0$ represents the charge transfer resistance of the electrode surface under conditions without TA. $R_{ct}$ represents the charge transfer resistance of the electrode surface under different TA concentrations.

This experiment used CS350H (Wuhan KST Instruments Co., Wuhan, Hubei Province, China) for EIS testing. Before the experiment began, the samples were first soaked in the solution for 0.5 h to stabilize the OCP. For samples continuously immersed for 0 h, 1 h, 3 h, 6 h, 10 h, 15 h, and 20 h, EIS detection was performed separately. In a stable OCP state, the automatic detection frequency range was $10^{-2} \sim 10^5$ Hz, and the signal amplitude disturbance was 10 mV. ZView software (3.10, AMETEK SCIENTIFIC INSTRUMENTS Co., Ltd., Poli, PA, USA) was used for fitting and impedance analysis.

*3.4. Microstructure Observation*

Upon subjecting the samples to a 20 h immersion in solutions of varying conditions, we employed ultrasonic cleaning techniques for sample decontamination, followed by a drying process to prepare the specimens for further analyses. Subsequent microstructural and compositional investigations were conducted utilizing a QUANTA 200 scanning electron microscope (SEM) integrated with energy-dispersive spectroscopy (EDS) manufactured by FEI Company (Hillsboro, OR, USA). During the SEM examination, operational parameters were set at a chamber pressure of 0.83 Torr and an acceleration voltage of 15.00 kV, with a magnification factor of 2000×. For EDS analysis, the assessments were performed under vacuum conditions, and the energy range was designated as 0.00–10.00 keV.

Additionally, X-ray photoelectron spectroscopy (XPS) analyses were executed using a Thermo Fisher Scientific instrument (Shanghai, China). The excitation radiation was chosen to be 1486.6 eV with a spot size of 380 μm. Constant pass energies were set at 100 eV for comprehensive sample examination and 50 eV for individual elemental analysis. The

energy step size was adjusted to 0.100 eV. All the acquired XPS data were processed and fitted using XPSPEAK 4.1 software.

## 4. Results and Discussion

### 4.1. Weight Loss Method

The average mass of the W18Cr4V specimens was meticulously measured before and after a 20 h immersion period, and the average mass loss was subsequently calculated. Following this, corrosion rates ($v_r$) and corrosion efficiencies ($\eta_W$) were determined utilizing Formulas (1) and (2). These key parameters, vital for quantifying the corrosion phenomena, are comprehensively tabulated in Table 2. This quantitative analysis provides foundational data for the subsequent interpretation of corrosion mechanisms.

**Table 2.** Summary of weight loss method data for W18Cr4V after soaking for 20 h in solutions with different concentrations of TA.

| Concentration | Before Immersion (g) | After 20 h Immersion (g) | $\Delta W$ (g) | $v_r$ (g(cm$^2$h)) | $\eta_W$ (%) |
|---|---|---|---|---|---|
| 0 wt% TA | $7.634 \pm 0.01$ | $7.508 \pm 0.002$ | 0.126 | 649.27 | - |
| 1 wt% TA | $7.223 \pm 0.01$ | $7.083 \pm 0.002$ | 0.140 | 722.31 | $-11.25$ |
| 3 wt% TA | $7.461 \pm 0.01$ | $7.315 \pm 0.002$ | 0.146 | 753.41 | $-16.04$ |
| 5 wt% TA | $7.463 \pm 0.01$ | $7.382 \pm 0.002$ | 0.081 | 418.65 | 35.52 |
| 8 wt% TA | $6.412 \pm 0.01$ | $6.357 \pm 0.002$ | 0.055 | 282.63 | 56.47 |
| 10 wt% TA | $7.595 \pm 0.01$ | $7.550 \pm 0.002$ | 0.045 | 233.09 | 64.11 |

Upon meticulous analysis of the data presented in Table 2, we observed a significant dependency between the concentrations of tannic acid and the corrosion rates of W18Cr4V. Specifically, at tannic acid concentrations of 1 wt% and 3 wt%, the $v_r$ in a 10 wt% acetic acid solution increased substantially relative to the 0 wt% baseline, the corresponding $\eta_W$ were $-11.25\%$ and $-16.04\%$, respectively. These negative values imply that tannic acid, within this concentration range, may act as a corrosion promoter for W18Cr4V. Conversely, when the tannic acid concentration escalated to 5 wt%, the $v_r$ experienced a notable decrease relative to the 0 wt% baseline and $\eta_W$ turned positive, reaching 35.52%. This inflection point suggests that tannic acid begins to serve as a corrosion inhibitor at this concentration. Upon further increase to 8 wt% and 10 wt% concentrations, the $v_r$ substantially declined, and the corresponding $\eta_W$ were 56.47% and 64.11%, respectively. This trend strongly indicates that in a 10 wt% acetic acid environment, elevated concentrations of tannic acid (particularly at 8 wt% and 10 wt%) act as significant corrosion inhibitors for W18Cr4V, and the inhibitive effect is positively correlated with the concentration of tannic acid.

### 4.2. Electrochemical Testing

#### 4.2.1. PDP Test

The polarization curve of W18Cr4V samples that were continuously immersed in acetic acid solution with different concentrations of tannic acid for 20 h is shown in Figure 1. After fitting the polarization curve, the corrosion potential ($E_{corr}$), corrosion current density ($I_{corr}$), anodic Tafel constant ($\beta_a$), and cathodic Tafel constant ($\beta_c$) are obtained. The corrosion or inhibition efficiency ($\eta_P$) is calculated using Formula (3), and the specific results are listed in Table 3.

By analyzing Figure 1, it can be observed that as the concentration of tannic acid increased, $E_{corr}$ of W18Cr4V in the solution gradually shifted to the negative side. The anodic slope decreased with the increase in tannic acid concentration. The trend of the cathodic slope was to increase slightly first and then decrease significantly, with significant changes starting at 5 wt% TA. Additionally, the shift in the cathodic part of the polarization curve was more pronounced, while the shift in the anodic part was relatively uniform.

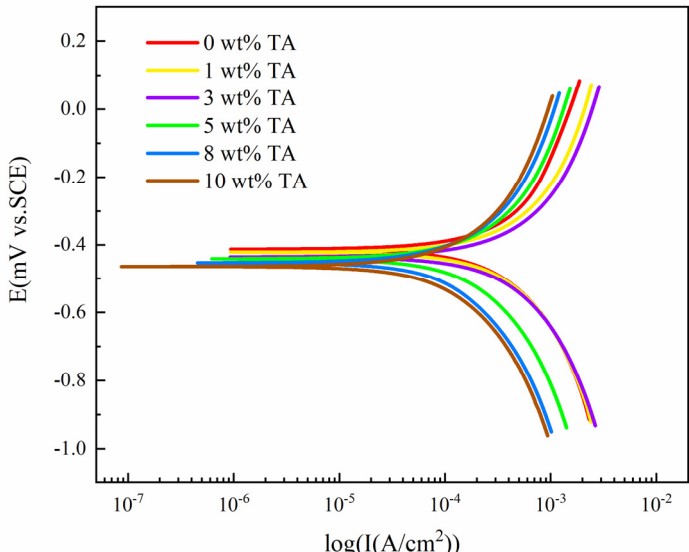

**Figure 1.** Polarization curve for W18Cr4V after soaking for 20 h in solutions with different concentrations of TA.

**Table 3.** Summary of polarization curve data for W18Cr4V after soaking for 20 h in solutions with different concentrations of TA.

| Concentration | $E_{corr}$ (mV vs. SCE) | $I_{corr}$ (μA/cm$^2$) | $\beta_a$ (mV/dec) | $\beta_c$ (mV/dec) | $\eta_P$ (%) |
|---|---|---|---|---|---|
| 0 wt% TA | $-4.16 \times 10^2$ | 106.92 | 173.51 | 201.89 | - |
| 1 wt% TA | $-4.24 \times 10^2$ | 118.92 | 165.77 | 202.02 | $-11.22$ |
| 3 wt% TA | $-4.37 \times 10^2$ | 125.05 | 161.40 | 204.58 | $-16.01$ |
| 5 wt% TA | $-4.40 \times 10^2$ | 69.12 | 153.53 | 185.63 | 35.49 |
| 8 wt% TA | $-4.56 \times 10^2$ | 46.59 | 155.37 | 175.62 | 56.43 |
| 10 wt% TA | $-4.67 \times 10^2$ | 38.42 | 154.72 | 170.15 | 64.06 |

These results indicate that the addition of tannic acid has a significant impact on the polarization behavior of W18Cr4V in acetic acid solution, especially in the cathodic part. Therefore, tannic acid can be considered a cathodic inhibitor.

In Table 3, as the concentration of tannic acid increased, $I_{corr}$ of W18Cr4V in the solution first increased slightly and then decreased significantly. When the concentration of tannic acid reached 5 wt%, it began to decrease significantly. When the concentration of tannic acid rose to 10 wt%, it was less than one-third of that of the 3 wt% tannic acid. Moreover, $\eta_W$ obtained from WL and $\eta_P$ from PDP showed high consistency, further confirming that lower concentrations of tannic acid (1 wt%, 3 wt%) promote corrosion of W18Cr4V in 10 wt% AA, while higher concentrations (5 wt%, 8 wt%, 10 wt%) have a significant corrosion inhibitory effect on W18Cr4V.

### 4.2.2. EIS Test

Figure 2 displays the impedance spectra of W18Cr4V in tannic acid solutions of different concentrations. The semi-circular impedance in the high-frequency region represents the electrode surface impedance $R_f$, while the semi-circular impedance in the mid-low frequency region reflects the charge transfer impedance $R_{ct}$ [43]. In the ultra-low frequency region, the appearance of inductance in the solution without tannic acid is attributed to the relaxation effect caused by the adsorption of the intermediate product $Fe(CH_3COO)_2$ on the metal surface [44]. In solutions containing tannic acid, the appearance of the inductive loop is due to the adsorption and desorption process of tannic acid [45,46]. In Figure 2a–c, as the immersion time increases, the capacitance arc decreases with the corresponding impedance value, mainly because cracks appear on the surface of W18Cr4V during immersion, increasing the contact area with the solution and accelerating the corrosion process. However, in

Figure 2d–f, the capacitance arc decreases first with immersion time and then increases or continues to increase, with the corresponding impedance value showing the same trend. This is mainly because high concentrations of tannic acid (5 wt%, 8 wt%, 10 wt%) have a good inhibitory effect, forming an effective iron–tannin complex on the substrate surface, isolating the substrate from the solution, thereby preventing corrosion

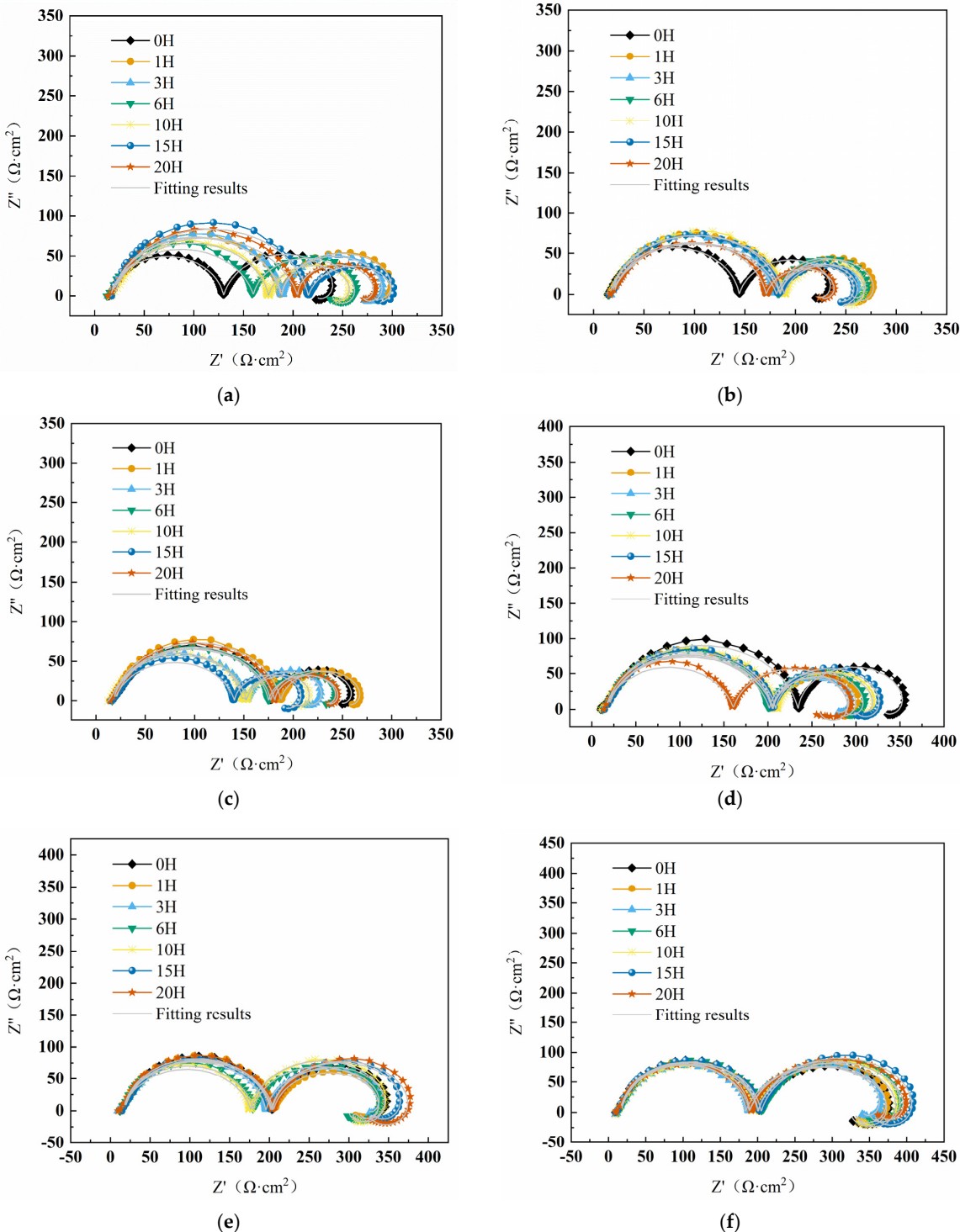

**Figure 2.** Impedance spectral curves of W18Cr4V after soaking for 20 h in solutions with different concentrations of TA (0 wt% TA (**a**), 1 wt% TA (**b**), 3 wt% TA (**c**), 5 wt% TA (**d**), 8 wt% TA (**e**), 10 wt% TA (**f**)).

An equivalent circuit model $R_S\left(C_f\left(R_f(C_{dl}R_{ct}R_LL)\right)\right)$ was used to fit the impedance spectrum curve, and the fitting circuit is shown in Figure 3. In the model, $R_s$ represents the resistance of the solution, $C_f$ represents the capacitance between the corrosion product film and the solution, $R_f$ represents the resistance of the corrosion product film, $C_{dl}$ represents the capacitance between the corrosion product film and the substrate, $R_{ct}$ represents the electron transfer resistance, $R_L$ represents the electron migration resistance in the corrosion nucleation area, $L$ is the inductance, and $L$ and $R_L$ together describe the surface corrosion reaction. The corrosion or inhibition efficiency $\eta_E$ were calculated using Formula (4), and the fitting data are summarized in Table 4.

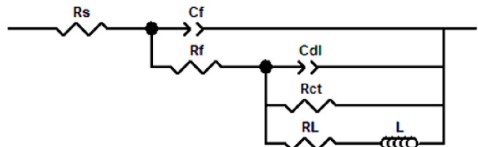

**Figure 3.** Equivalent circuit of the impedance curve of W18Cr4V.

The formula for calculating the double-layer capacitance is shown in Formula (5).

$$C_{dl} = 1/2\pi f_{max}R_{ct}, \tag{5}$$

where $f_{max}$ represents the frequency value of the maximum imaginary part of the impedance.

By analyzing the data in Table 4, we found that $L$ decreased with the increase in tannic acid concentration and the extension of immersion time. This is because the higher the concentration, the more iron–tannin complexes are formed and the stronger the adsorption of tannic acid molecules on the W18Cr4V surface. At low concentrations (1 wt%, 3 wt%), $C_{dl}$ gradually increased, while at higher concentrations (5 wt%, 8 wt%, 10 wt%), it gradually decreased, which is mainly related to the concentration of tannic acid [47]. Compared with the solution without tannic acid, 1 wt% and 3 wt% tannic acid did not show corrosion inhibition on W18Cr4V during the soaking period but promoted corrosion, with inhibition efficiencies of −11.15% and −15.75%, respectively. An amount of 5 wt% tannic acid began to show inhibitory effects on the samples after 6 h of continuous immersion, and the inhibition efficiency after 20 h of immersion was 35.57%. Meanwhile, 8 wt% and 10 wt% tannic acid showed good inhibitory effects on W18Cr4V, with inhibition efficiencies of 56.43% and 63.91%, respectively. The main reason is that at low concentrations (1 wt%, 3 wt%) of TA, the iron–tannin complex formed by tannic acid and iron is limited by the tannic acid concentration and cannot form a dense inhibitory layer on the W18Cr4V surface. Therefore, tannic acid and iron continuously undergo chelation reactions, indirectly promoting corrosion. At a 5 wt% concentration, the inhibitory effect began to appear after 6 h of continuous immersion, indicating that the amount of iron–tannin complex formed increased and more tannic acid molecules were adsorbed on the surface, which can slow down corrosion [48–50]. At concentrations of 8 wt% and 10 wt%, the inhibition efficiency increased significantly after 20 h of continuous immersion, indicating that the formed iron–tannin complex significantly increased, enough to build a dense and effective inhibitory layer, thereby exerting a significant inhibitory effect. The key step in corrosion inhibition is believed to be the adsorption of tannic acid molecules, replacing the pre-adsorbed water molecules [51].

The inhibition efficiency calculated by EIS is consistent with the results calculated by WL and PDP. If the inevitable errors in the normal fitting process are excluded, the differences between these three methods are minimal.

**Table 4.** Summary of data on the fitting result of different impedance spectral curves for W18Cr4V after soaking for 20 h in solutions with different concentrations of TA.

| Concentration | Time (h) | $R_s$ (Ω cm$^2$) | $R_f$ (Ω cm$^2$) | $Y_{dl}$ (Sn/(Ω cm$^2$))10$^{-4}$ | $n_{dl}$ | $f_{max}$ (Hz) | $R_{ct}$ (Ω cm$^2$) | $C_{dl}$ (F/cm$^2$) | $R_L$ (Ω cm$^2$) | $L$ (H) | $\eta_E$ (%) |
|---|---|---|---|---|---|---|---|---|---|---|---|
| | 0 | 16.20 | 113.6 | 52.11 | 0.872 | 0.264 | 130.2 | 46.29 | 336.4 | 1327 | - |
| | 1 | 17.45 | 170.1 | 47.82 | 0.899 | 0.264 | 127.1 | 47.42 | 360 | 1564 | - |
| | 3 | 17.60 | 171.2 | 35.63 | 0.946 | 0.423 | 107.9 | 34.89 | 332.5 | 1611 | - |
| 0 wt% TA | 6 | 15.21 | 144.9 | 11.52 | 0.918 | 0.264 | 107.2 | 56.22 | 310.7 | 1871 | - |
| | 10 | 20.1 | 155.2 | 107.41 | 0.882 | 0.165 | 99.24 | 97.08 | 221.6 | 2121 | - |
| | 15 | 18.32 | 197.4 | 150.27 | 0.848 | 0.103 | 96.68 | 159.21 | 176.4 | 2708 | - |
| | 20 | 17.2 | 187.5 | 77.83 | 0.985 | 0.103 | 80.34 | 191.59 | 300.4 | 2214 | - |
| | 0 | 15.25 | 190.7 | 105.78 | 0.928 | 0.210 | 117.9 | 64.44 | 204.7 | 2530 | −10.43 |
| | 1 | 18.49 | 166.4 | 79.03 | 0.890 | 0.210 | 112.9 | 67.29 | 248.5 | 2290 | −12.58 |
| | 3 | 18.62 | 166.4 | 63.72 | 0.994 | 0.264 | 91.34 | 65.98 | 361.8 | 2530 | −18.13 |
| 1 wt% TA | 6 | 17.67 | 163.9 | 210.50 | 0.890 | 0.264 | 88.6 | 68.02 | 183.2 | 2050 | −20.99 |
| | 10 | 20.03 | 151.1 | 153.26 | 0.853 | 0.210 | 90.74 | 83.72 | 162.4 | 2140 | −9.37 |
| | 15 | 18.61 | 166.6 | 140.55 | 0.914 | 0.210 | 79.45 | 95.62 | 148.1 | 2350 | −21.69 |
| | 20 | 16.23 | 211.5 | 151.70 | 0.988 | 0.210 | 72.28 | 105.10 | 152.4 | 1341 | −11.15 |
| | 0 | 20.12 | 157.1 | 106.53 | 0.880 | 0.165 | 93.01 | 103.58 | 275.1 | 2349 | −39.99 |
| | 1 | 17.48 | 169.9 | 118.37 | 0.897 | 0.131 | 88.79 | 137.03 | 240.4 | 2874 | −43.15 |
| | 3 | 17.82 | 134.7 | 116.73 | 0.893 | 0.165 | 87.62 | 109.95 | 189.9 | 1569 | −23.15 |
| 3 wt% TA | 6 | 20.11 | 155 | 147.28 | 0.899 | 0.131 | 76.78 | 158.46 | 208.8 | 2049 | −39.62 |
| | 10 | 14.51 | 137.1 | 161.89 | 0.892 | 0.131 | 74.09 | 164.22 | 128.1 | 1406 | −33.95 |
| | 15 | 18.78 | 121.7 | 147.12 | 0.924 | 0.131 | 73.86 | 164.73 | 112.9 | 1688 | −30.90 |
| | 20 | 17.33 | 164.1 | 160.99 | 0.948 | 0.131 | 69.41 | 175.29 | 191.8 | 2258 | −15.75 |
| | 0 | 15.75 | 218.7 | 43.29 | 0.954 | 0.264 | 128.1 | 47.05 | 394.7 | 2181 | −1.64 |
| | 1 | 16.43 | 127.9 | 80.59 | 0.862 | 0.131 | 109.2 | 111.42 | 269.3 | 1587 | −16.39 |
| | 3 | 16.66 | 188.5 | 45.51 | 0.962 | 0.423 | 93.07 | 40.43 | 321.5 | 1429 | −15.93 |
| 5 wt% TA | 6 | 16.19 | 185.6 | 24.32 | 0.935 | 0.674 | 114.1 | 20.71 | 340.8 | 1220 | 6.05 |
| | 10 | 15.61 | 193.4 | 57.12 | 0.925 | 0.264 | 117.1 | 51.47 | 532.8 | 1287 | 15.25 |
| | 15 | 18.44 | 187.7 | 18.88 | 0.928 | 0.210 | 128.2 | 59.26 | 353.9 | 1345 | 24.59 |
| | 20 | 15.23 | 145.7 | 5.85 | 0.889 | 0.264 | 124.7 | 44.74 | 352.5 | 1052 | 35.57 |

**Table 4.** *Cont.*

| Concentration | Time (h) | $R_s$ ($\Omega$ cm$^2$) | $R_f$ ($\Omega$ cm$^2$) | $Y_{dl}$ (Sn/($\Omega$ cm$^2$))10$^{-4}$ | $n_{dl}$ | $f_{max}$ (Hz) | $R_{ct}$ ($\Omega$ cm$^2$) | $C_{dl}$ (F/cm$^2$) | $R_L$ ($\Omega$ cm$^2$) | $L$ (H) | $\eta_E$ (%) |
|---|---|---|---|---|---|---|---|---|---|---|---|
| | 0 | 16.12 | 187.7 | 23.65 | 0.933 | 0.423 | 153.4 | 24.53 | 377.7 | 1616 | 15.12 |
| | 1 | 16.66 | 188.6 | 12.46 | 0.892 | 1.071 | 148 | 10.04 | 378.4 | 1074 | 14.12 |
| | 3 | 16.06 | 179.6 | 28.89 | 0.889 | 0.423 | 158.5 | 23.74 | 717.5 | 1210 | 31.92 |
| 8 wt% TA | 6 | 16.06 | 163.5 | 6.30 | 0.887 | 1.718 | 166.3 | 5.57 | 438.7 | 1137 | 35.54 |
| | 10 | 15.55 | 161.1 | 12.35 | 0.914 | 0.853 | 180.6 | 10.34 | 428.8 | 1537 | 45.05 |
| | 15 | 17.69 | 184.3 | 20.41 | 0.894 | 0.531 | 181 | 16.57 | 433.9 | 1661 | 46.59 |
| | 20 | 16.07 | 186.2 | 11.01 | 0.896 | 0.852 | 184.4 | 10.13 | 364.7 | 1360 | 56.43 |
| | 0 | 15.85 | 184.5 | 7.08 | 0.882 | 1.718 | 182.2 | 5.09 | 398.8 | 1400 | 28.54 |
| | 1 | 15.9 | 184.3 | 12.76 | 0.879 | 0.852 | 188.1 | 9.93 | 426.5 | 1381 | 32.43 |
| | 3 | 17.74 | 165.8 | 16.79 | 0.818 | 0.674 | 206.3 | 11.45 | 701 | 1478 | 47.70 |
| 10 wt% TA | 6 | 17.32 | 187.4 | 15.50 | 0.867 | 0.674 | 208.1 | 11.35 | 483 | 1218 | 48.49 |
| | 10 | 17.98 | 171 | 7.47 | 0.836 | 1.355 | 215.1 | 5.46 | 427.5 | 1098 | 53.86 |
| | 15 | 15.83 | 184.9 | 9.30 | 0.875 | 0.852 | 222.4 | 8.40 | 523.6 | 1145 | 56.53 |
| | 20 | 15.58 | 173 | 9.58 | 0.839 | 1.071 | 222.6 | 6.67 | 840.9 | 1170 | 63.91 |

### 4.3. Surface Analysis

4.3.1. SEM and EDS Analysis

Figure 4 demonstrates the SEM and EDS analysis results of W18Cr4V after soaked for 20 h under various solution conditions. As can be clearly observed in the figure, under corrosion conditions with tannic acid concentrations of 0 wt%, 1 wt%, and 3 wt% (Figure 4b–d), compared to the unsoaked specimens (Figure 4a), W18Cr4V experienced significant corrosion in these three solutions, forming corrosion cracks of varying sizes. Moreover, the depth of the cracks slightly increased with the rise in tannic acid concentration. At a tannic acid concentration of 5 wt% (Figure 4e), the condition of W18Cr4V notably improved, revealing a smoother surface. When the concentration of tannic acid was 8 wt% and 10 wt% (Figure 4f,g), the corrosion inhibition effect was prominent, with a tightly bound layer of iron–tannic acid complex clearly observed on the surface of W18Cr4V.

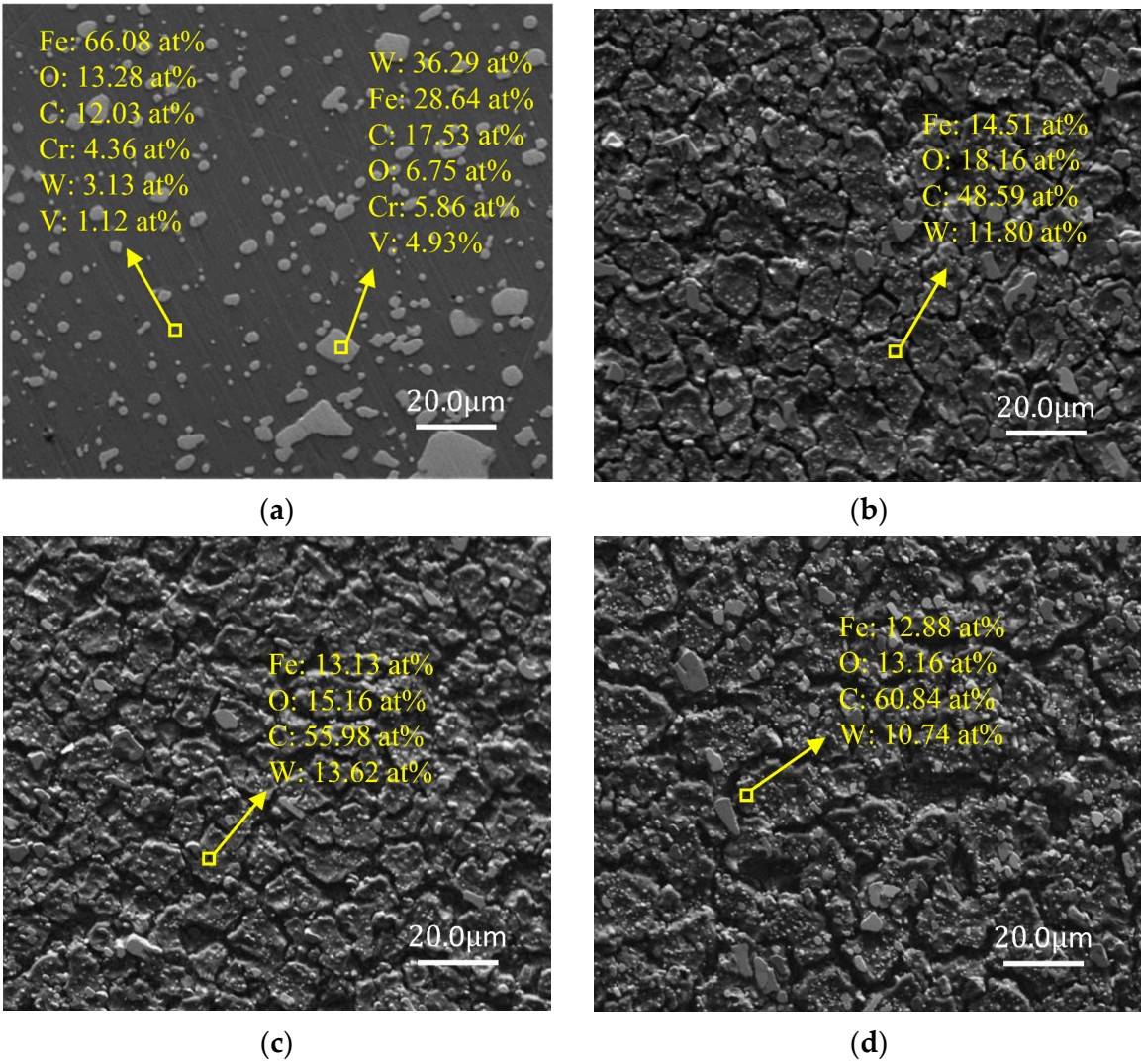

**Figure 4.** *Cont.*

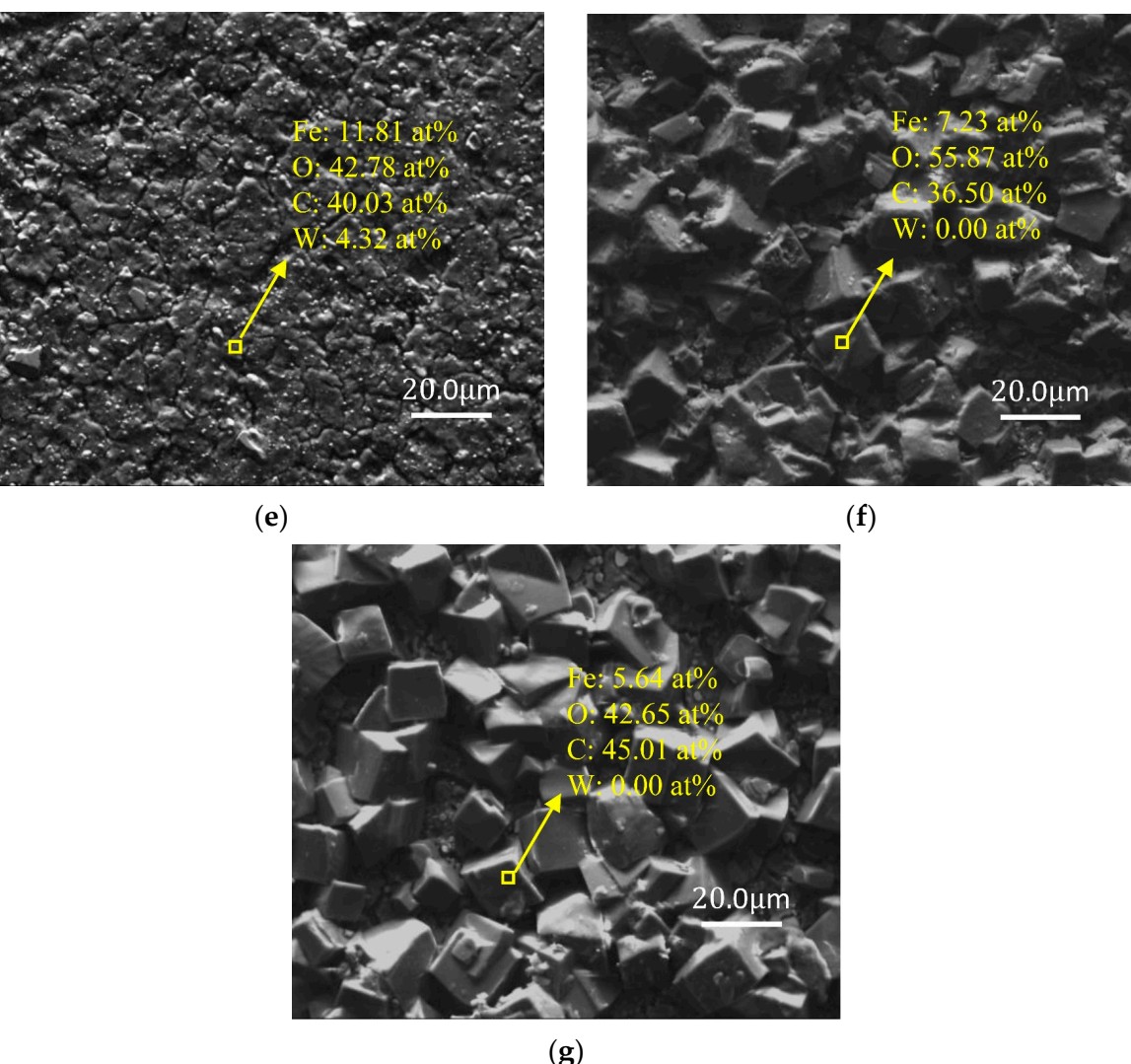

**Figure 4.** SEM and EDS images of W18Cr4V after treatment under different conditions for 20 h (unprocessed (**a**), 0 wt% TA (**b**), 1 wt% TA (**c**), 3 wt% TA (**d**), 5 wt% TA (**e**), 8 wt% TA (**f**), 10 wt% TA (**g**)).

The EDS analysis results in Figure 4 reveal that after 20 h of immersion, in the absence of tannic acid (0 wt% TA), iron was stripped from W18Cr4V due to the reaction with acetic acid, resulting in the bare specimen matrix, i.e., the percentage of W and C increased (Figure 4b). In the case of lower concentrations of tannic acid (1 wt% TA, 3 wt% TA), a rather small amount of iron–tannin complex formed. Such a small amount of corrosion product on the surface of the specimen did not show a corrosion inhibition effect, and chelation accelerated the corrosion of W18Cr4V in the acetic acid environment, resulting in a further increase in the content of C and W (Figure 4c,d). When the concentration of tannic acid was increased to 5 wt%, the proportion of the complexes in the solution gradually increased, and complexes began to cover the surface of the specimen, forming an initial corrosion inhibition layer, resulting in a decrease in the proportion of W and an increase in the proportion of O (Figure 4e). Finally, when the tannic acid concentration reached 8 wt% and 10 wt%, this dense complex completely covered the surface of the specimen, resulting in the EDS no longer detecting the presence of W (Figure 4f,g). This corrosion product layer primarily forms through chemical adsorption, and its adsorbed layer is even visible to the naked eye. It remains intact even after ultrasonic cleaning, indicating that the adsorption layer is highly stable and tightly bound, providing protection to W18Cr4V.

In addition, the SEM and EDS measurement results further confirm the results of WL, PDP, and EIS.

### 4.3.2. XPS Spectrum Analysis

Figures 5–8 show the XPS scanning diagrams and detailed spectra of some elements of the surface products of W18Cr4V after soaking for 20 h under different conditions. Observation of the plots reveals that the peaks of C and O were more prominently exhibited under the conditions of 3 wt% TA and 10 wt% TA compared to the conditions of 0 wt% TA. C and O are the basic components of the reaction products, so it can be proved that the iron–tannic acid complexes were adsorbed on the surface of W18Cr4V after the addition of tannic acid. Additionally, the content of iron and its oxides decreased in the conditions of 3 wt% and 10 wt% compared to those in the conditions of 0 wt% TA, which also indicated that the corrosion products were covered on the surface of W18Cr4V. The peaks of W and Cr existed in the environment of 0 wt% TA, but the intensities of these two elements decreased in other conditions, and this phenomenon further confirmed that the corrosion products were covered on the surface of the specimen. The binding energies of the XPS peaks and their possible corresponding compounds are shown in Table 5.

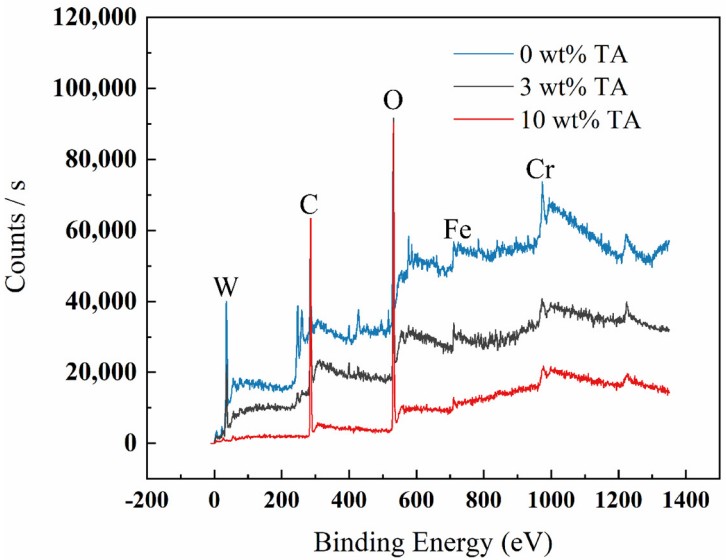

**Figure 5.** XPS spectral scans of surface products of W18Cr4V after immersion for 20 h under different conditions.

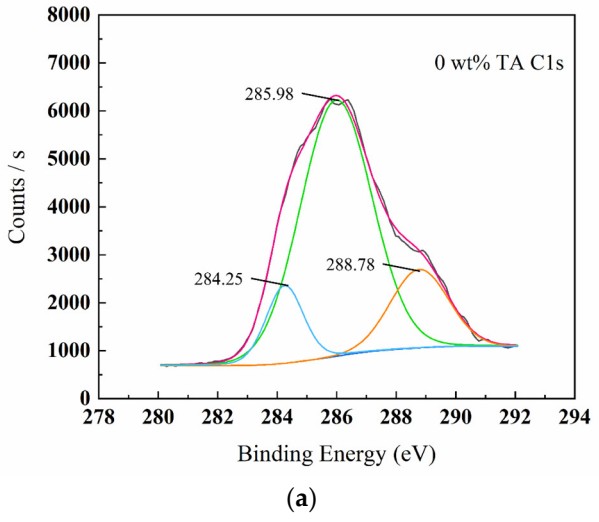

(a)

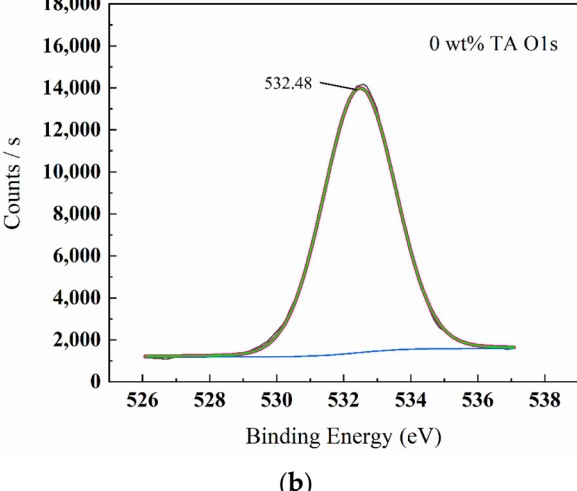

(b)

**Figure 6.** *Cont*.

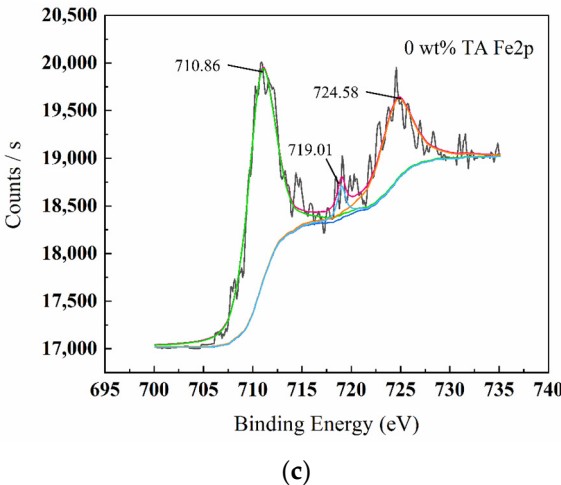

(**c**)

**Figure 6.** XPS spectrum graphs of surface products C1s (**a**), O1s (**b**), Fe2p (**c**) for W18Cr4V after treatment in 0 wt% TA for 20 h.

(**a**)

(**b**)

(**c**)

**Figure 7.** XPS spectrum graphs of surface products C1s (**a**), O1s (**b**), Fe2p (**c**) for W18Cr4V after treatment in 3 wt% TA for 20 h.

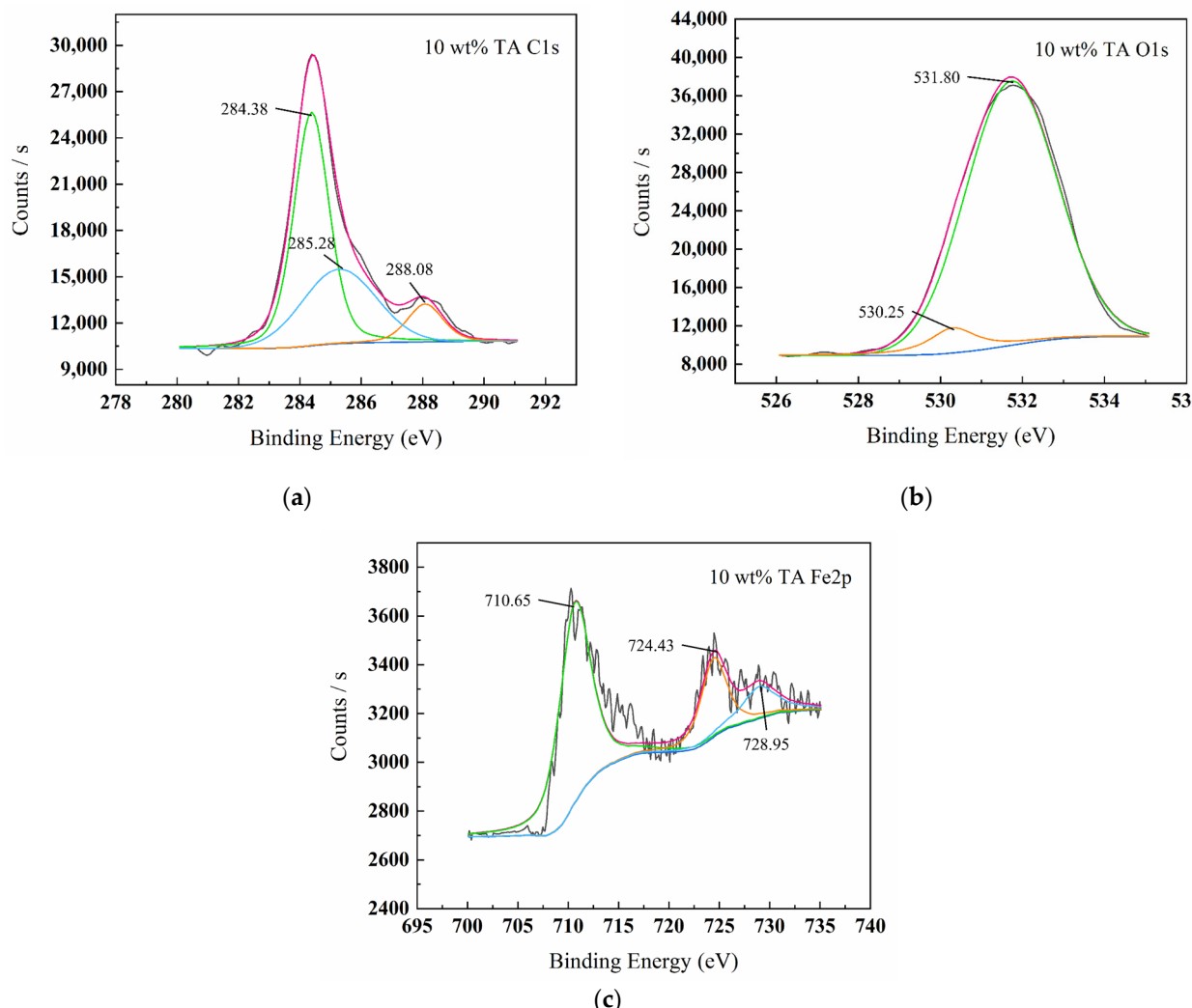

**Figure 8.** XPS spectrum graphs of surface products C1s (**a**), O1s (**b**), Fe2p (**c**) for W18Cr4V after treatment in 10 wt% TA for 20 h.

**Table 5.** Summary of XPS spectral analysis of surface products of W18Cr4V after soaking for 20 h in solutions with different concentrations of TA.

| Concentration | Peak | Energy (eV) | Possible Assignment | References |
|---|---|---|---|---|
| 0 wt% TA | Fe2p | 710.86 | $Fe(CH_3COO)_2$, $Fe(OH)_3$, FeO, $Fe_2O_3$ | [52] |
| | | 719.01 | $Fe(CH_3COO)_2$, $Fe(OH)_3$, $Fe_3O_4$, $Fe_2O_3$ | [52] |
| | | 724.58 | $Fe(CH_3COO)_2$, $Fe(OH)_3$, $Fe_3O_4$, $Fe_2O_3$ | [53] |
| | O1s | 532.48 | C=O | [53] |
| | C1s | 284.25 | C−C or C=C | [54] |
| | | 285.98 | C=C or C=O | [54] |
| | | 288.78 | O−C=O | [53] |
| 3 wt% TA | Fe2p | 711.38 | Fe(III)−tannates, $Fe(CH_3COO)_2$, $Fe(OH)_3$, FeO, $Fe_2O_3$ | [55] |
| | | 715.74 | Fe(III)−tannates, $Fe(CH_3COO)_2$, $Fe(OH)_3$, $Fe_2O_3$ | [55] |
| | | 718.03 | Fe(III)−tannates, $Fe(CH_3COO)_2$, $Fe(OH)_3$, $Fe_3O_4$, $Fe_2O_3$ | [55] |
| | | 725.50 | Fe(III)−tannates, $Fe(CH_3COO)_2$, $Fe(OH)_3$, $Fe_3O_4$, $Fe_2O_3$ | [55] |
| | O1s | 531.08 | Fe−O | [56] |
| | | 532.48 | C=O | [53] |
| | C1s | 284.58 | C−C or C=C | [54] |
| | | 286.21 | C=C or C=O | [54] |
| | | 288.11 | O−C=O | [53] |

**Table 5.** *Cont.*

| Concentration | Peak | Energy (eV) | Possible Assignment | References |
|---|---|---|---|---|
| 10 wt% TA | Fe2p | 710.65 | Fe(III)−tannates, Fe(CH$_3$COO)$_2$, Fe(OH)$_3$, FeO, Fe$_2$O$_3$ | [55] |
| | | 724.43 | Fe(III)−tannates, Fe(CH$_3$COO)$_2$, Fe(OH)$_3$, Fe$_3$O$_4$, Fe$_2$O$_3$ | [55] |
| | | 782.95 | Fe(III)−tannates, Fe(CH$_3$COO)$_2$, Fe(OH)$_3$, Fe$_3$O$_4$, Fe$_2$O$_3$ | [55] |
| | O1s | 530.25 | Fe−O | [56] |
| | | 531.80 | C=O | [53] |
| | C1s | 284.38 | C−C or C=C | [54] |
| | | 285.28 | C=C or C=O | [54] |
| | | 288.08 | O−C=O | [53] |

By combining the various detection methods and surface analysis results, we clarified that by adding different concentrations of tannic acid under the condition of 10 wt% acetic acid, the iron in W18Cr4V reacts with tannic acid to form iron–tannic acid complexes. At low concentrations (1 wt%, 3 wt%) of TA, the complex content is small, not enough to form a corrosion inhibition layer on the surface of W18Cr4V, but instead indirectly promotes corrosion due to the continuous chelating effect. At high TA concentrations (5 wt%, 8 wt%, 10 wt%), the complex content is large. The compound is adsorbed on the surface of W18Cr4V and thus forms a layer of corrosion inhibition film, which insulates the corrosive substances from contacting with the metal surface, inhibiting the corrosion of W18Cr4V steel in acetic acid.

## 5. Conclusions and Discussion

This study delves deeply into the corrosion behavior of W18Cr4V material in a simulated wood environment. Based on the experimental data and analysis results, the following conclusions can be drawn:

1. The corrosion mechanism of W18Cr4V in a simulated wood environment significantly depends on the concentration of tannic acid. Low concentrations of tannic acid promote corrosion, whereas high concentrations exhibit a strong corrosion inhibition effect. This finding reveals the complex regulatory role of tannic acid concentration on metal corrosion.

2. Weight loss method (WL), potentiodynamic polarization (PDP), and electrochemical impedance spectroscopy (EIS) provided compelling empirical evidence for this study, confirming the influence of tannic acid concentration on the corrosion properties of W18Cr4V. Tannic acid acts as a cathodic inhibitor for W18Cr4V in the simulated wood environment. After 20 h of continuous immersion, compared to the 0 wt% TA solution, the corrosion inhibition efficiency of the 1 wt% TA and 3 wt% TA solutions are −11.2% and −16.0%, respectively. However, in the 5 wt% tannic acid solution, a noticeable corrosion inhibition effect can be observed from 6 h of immersion, with an inhibition efficiency of 35.6% after 20 h. For the 8 wt% and 10 wt% tannic acid solutions, their corrosion inhibition efficiencies reached 56.4% and 64.1%, respectively. This discovery reveals the unique value of high concentrations (8 wt%, 10 wt%) of tannic acid for metal corrosion protection.

3. Analytical methods such as scanning electron microscopy (SEM), energy-dispersive X-ray spectroscopy (EDS), and X-ray photoelectron spectroscopy (XPS) confirmed the microstructure and composition of the surface film of W18Cr4V material. At low concentrations (1 wt%, 3 wt%), after 20 h of continuous immersion, tannic acid indirectly accelerated the corrosion process of W18Cr4V due to its persistent chelation action. When the concentration of tannic acid increased (8 wt%, 10 wt%), an iron–tannic acid complex inhibitive film was formed on the surface of the W18Cr4V material, effectively slowing down the corrosion process. This conclusion provides key information on the microscopic mechanism of the interaction between tannic acid and the metal interface, further supporting the aforementioned inhibition mechanism.

In this study, we conducted a comprehensive investigation into the effects of tannic acid on the corrosion behavior of woodworking tool material W18Cr4V in a simulated wood environment based on a 10 wt% acetic acid solution. However, to fully understand this phenomenon and translate it into practical applications, several directions warrant further exploration.

First and foremost, the accuracy of environmental simulation is a critical factor. The 10 wt% acetic acid solution employed in the current study requires further validation for its suitability as a representative wood environment. This would involve the incorporation of parameters that more closely emulate real-world wood conditions, as well as the possible interaction effects among multiple factors.

Beyond short-term corrosion behavior, the long-term stability and potential side effects also constitute important avenues for future research. This encompasses the evaluation of the sustained efficacy of tannic acid in inhibiting W18Cr4V corrosion over extended periods, as well as the identification of any adverse effects. Building upon this, the development of novel eco-friendly corrosion inhibitors or organic coatings based on tannic acid presents an attractive application prospect. Such advances are not confined to woodworking tools but could be extended to other applications within the broader metal corrosion domain.

Furthermore, practical application tests, as well as economic and sustainability assessments, are also critical. Conducting large-scale experiments in actual woodworking environments and comparing the cost-effectiveness and environmental impact against existing corrosion mitigation strategies will aid in evaluating the real-world applicability of these approaches. Additionally, the corrosion-inhibiting effects of tannic acid on other metal materials, such as carbon steel and stainless steel, as well as interdisciplinary research and data modeling and forecasting, represent directions worthy of attention.

In summary, although the current study offers preliminary insights into the effects of tannic acid on the corrosion behavior of W18Cr4V, multiple avenues still require in-depth investigation. Such research is essential for a comprehensive understanding of the phenomenon and for extending its applicability to a broader range of domains.

### 6. Patents

During the work reported in this manuscript, the patent "A kind of adjustable four-electrode electrochemical corrosion experimental device" was produced.

**Author Contributions:** Conceptualization, N.J. and C.W.; methodology, C.W.; software, C.Z.; validation, N.J. and J.L.; formal analysis, N.J.; investigation, C.Z.; resources, J.L.; data curation, C.W.; writing—original draft preparation, C.W.; writing—review and editing, N.J.; supervision, C.Z.; project administration, J.L.; funding acquisition, N.J. All authors have read and agreed to the published version of the manuscript.

**Funding:** This study was funded by the National Key Research and Development Program of the Ministry of Science and Technology of the State Council (Project No. '2022YFD2202105').

**Data Availability Statement:** The data that support the findings of this study are available on request from the corresponding author. The data are not publicly available due to privacy or ethical restrictions.

**Acknowledgments:** The authors would like to thank the anonymous reviewers and editorial staff for their comments, which greatly improved this manuscript.

**Conflicts of Interest:** The authors declare no conflict of interest.

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
