# Peer review of "Effect of Tannic Acid on the Corrosion Behavior of W18Cr4V in a Simulated Wood Environment and Its Inhibition Mechanism"

_forests, doi:10.3390/f14091781_

Round 1

Reviewer 1 Report

The aim of the paper, the main contributions and strengths and drawbacks:

1.       The authors described the effect of tannic acid on the corrosion of woodworking tool material. The electrochemical approach gives consistency to their hypotheses.

2.       The manuscript is clear, well-written, relevant to the field, presented in a well-structured manner, and the data is well-organized and explained.

3.       The manuscript fits the scope of the journal.

4.       Minor revisions detailed below are necessary:

Introduction section

5.       The references could be improved by adding at least 3 papers published in the last 3 years.

Materials and methods

6.       Additional data regarding the materials used for the experiments should be provided: manufacturer, characteristics, etc., for: tannic acid, acetic acid, the working electrode, etc.

7.       Explain why you chose the 10% wt. concentration for the acetic acid.

8.       Introduce details about the SEM-EDS survey (manufacturer, settings utilized when running the experiments, magnification, etc.)

      Data in Table 4 and Table 5 could be introduced in a supplementary materials file.

Results and discussions sections should be merged and there should exist at the end of the manuscript a Conclusions section.

A graphical abstract highlighting the main findings of this study could be also introduced to improve the design of this paper.

Minor editing of English language required.

Reviewer 2 Report

This is an interesting paper and the authors did a great job in presenting it in an informative and concise manner. Some minor revisions are required.

The introduction part and research background are clearly and well-written. However, it is not sure whether they can separate into two sections or should they combine under one Introduction section. Please check the journal’s house style.

Section 3.4 should be described in a more detailed manner.

The composition in Table 1 should be mentioned in full or put it in footnote.

The title of Table 2 did not reflect the data presented. Corrosion rate and efficiency of corrosion were not mentioned. Please revise.

Line 222 – what statistical results are presented? Only standard deviation is seen in Table 2

Figure 1 – can you fix the unit in the figure, superscript.

The results are presented in an organized and concise manner. However, more discussion is needed to strengthen the paper.

The conclusion is clear and met the objective.  

Author Response

请参阅附件。
